# State of Art of Idiosyncratic Drug-Induced Neutropenia or Agranulocytosis, with a Focus on Biotherapies

**DOI:** 10.3390/jcm8091351

**Published:** 2019-09-01

**Authors:** Emmanuel Andrès, Noel Lorenzo Villalba, Abrar-Ahmad Zulfiqar, Khalid Serraj, Rachel Mourot-Cottet, Jacques-Eric Gottenberg

**Affiliations:** 1Department of Internal Medicine, Medical Clinic B, University Hospital of Strasbourg, 67084 Strasbourg, France; 2Departments of Internal Medicine, University Hospital of Oujda, 59000 Oujda, Morocco; 3Department of Rheumatology, University Hospital of Strasbourg, 67084 Strasbourg, France; 4Referral Center of Immune Cytopenias, University Hospital of Strasbourg, 67084 Strasbourg, France

**Keywords:** drug, idiosyncratic, neutropenia, agranulocytosis, infections, antithyroid medications, ticlopidine, clozapine, sulfasalazine, antibiotics as trimethoprim-sulfamethoxazole (cotrimoxazole), and deferiprone, biotherapy, autoimmune disease, auto-inflammatory disorder, systemic vasculitis, orphan disease, anti-TNF-alpha agent, anti-CD20 agent, IL1-inhibitor, IL6 inhibitor, B-cell activating factor (BAFF) inhibitor

## Abstract

Introduction: Idiosyncratic drug-induced neutropenia and agranulocytosis is seldom discussed in the literature, especially for new drugs such as biotherapies outside the context of oncology. In the present paper, we report and discuss the clinical data and management of this relatively rare disorder, with a focus on biotherapies used in autoimmune and auto-inflammatory diseases. Materials and methods: A review of the literature was carried out using the PubMed database of the US National Library of Medicine. We searched for articles published between January 2010 and May 2019 using the following key words or associations: “drug-induced neutropenia”, “drug-induced agranulocytosis”, and “idiosyncratic agranulocytosis”. We included specific searches on several biotherapies used outside the context of oncology, including: tumor necrosis factor (TNF)-alpha inhibitors, anti-CD20 agents, anti-C52 agents, interleukin (IL) 6 inhibitors, IL 1 inhibitors, and B-cell activating factor inhibitor. Results: Idiosyncratic neutropenia remains a potentially serious adverse event due to the frequency of severe sepsis with severe deep tissue infections (e.g., pneumonia), septicemia, and septic shock in approximately two-thirds of all hospitalized patients with grade 3 or 4 neutropenia (neutrophil count (NC) ≤ 0.5 × 109/L and ≤ 0.1 × 109/L, respectively). Over the last 20 years, several drugs have been strongly associated with the occurrence of idiosyncratic neutropenia, including antithyroid drugs, ticlopidine, clozapine, sulfasalazine, antibiotics such as trimethoprim-sulfamethoxazole, and deferiprone. Transient grade 1–2 neutropenia (absolute blood NC between 1.5 and 0.5 × 109/L) related to biotherapy is relatively common with these drugs. An approximate 10% prevalence of such neutropenia has been reported with several of these biotherapies (e.g., TNF-alpha inhibitors, IL6 inhibitors, and anti-CD52 agents). Grade 3–4 neutropenia or agranulocytosis and clinical manifestations related to sepsis are less common, with only a few case reports to date for most biotherapies. Special mention should be made of late onset and potentially severe neutropenia, especially following anti-CD52 agent therapy. During drug therapy, several prognostic factors have been identified that may be helpful when identifying ‘susceptible’ patients. Older age (>65 years), septicemia or shock, renal failure, and a neutrophil count ≤0.1 × 109/L have been identified as poor prognostic factors. Idiosyncratic neutropenia should be managed depending on clinical severity, with permanent/transient discontinuation or a lower dose of the drug, switching from one drug to another of the same or another class, broad-spectrum antibiotics in cases of sepsis, and hematopoietic growth factors (particularly G-CSF). Conclusion: Significant progress has been made in recent years in the field of idiosyncratic drug-induced neutropenia, leading to an improvement in their prognosis (currently, mortality rate between 5 and 10%). Clinicians must continue their efforts to improve their knowledge of these adverse events with new drugs as biotherapies.

## 1. Introduction

In 1922, Schultz first introduced the term ‘agranulocytosis’ for cases of severe pharyngeal infections associated with a lack of granulocytes in the blood. Agranulocytosis is associated with a profound decrease (severe neutropenia) or a complete lack of the number of granulocytes in circulating blood, classically resulting in a neutrophil count of less than 0.5 × 109/L [1,2]. Patients with such severe neutropenia are likely to develop life-threatening and sometimes fatal infections. Since the original description, neutropenia, defined as an absolute neutrophil count of less than 1.5 × 109/L, has been related to most classes of medications (Table 1) [2,3]. For most drugs, the risk is likely to be very small. For medications such as antithyroid drugs, ticlopidine, clozapine, sulfasalazine, trimethoprim-sulfamethoxazole, and dipyrone, the risk may, however, be higher [2]. In the last 10 years, several reference papers have been published focused on drug-induced neutropenia and agranulocytosis [1,2,3,4]. However, none of these papers include specific data on biotherapies outside the context of oncology (e.g., infliximab, etanercept, rituximab, and tocilizumab), the latter being increasingly used particularly in autoimmune or auto-inflammatory diseases [5,6,7,8,9,10].

Therefore, we carried out a review on this topic, specifically focused on the biotherapies used in autoimmune and auto-inflammatory disorders.

## 2. Search Strategy

A literature search was performed via the PubMed database of the US National Library of Medicine. We searched for articles published between January 2010 and July 2019 using the following key words or associations: “drug-induced neutropenia”, “drug-induced agranulocytosis”, “idiosyncratic neutropenia”, and “idiosyncratic agranulocytosis”. Restrictions included: English-, Spanish-, or French-language publications; papers published from 1 January 2010 to 31 July 2019; human subjects; and clinical trials, review articles, or guidelines. We centered our research on several biotherapies on the market in USA and Europe, including: (i) tumor necrosis factor (TNF)-alpha (α) inhibitors: infliximab, adalimumab and etanercept; (ii) anti-CD20 agents: rituximab and obinutuzumab; (iii) anti-C52 agents: alemtuzumab; (iv) interleukin (IL) 6 inhibitors: tocilizumab; (v) IL1 inhibitors: anakinra and canakinumab; and (vi) B-cell activating factor (BAFF) inhibitor: belimumab. We restricted our research to the utilization of these drugs outside the context of oncology, in autoimmune and auto-inflammatory disorders, systemic vasculitis, or orphan diseases (e.g., rheumatoid arthritis, systemic lupus erythematous, Sjögren’s syndrome, Still’s disease, Behçet’s disease, giant cell arteritis, Crohn’s disease, ulcerative colitis, psoriatic arthritis, granulomatosis with polyangiitis, and genetic fevers) (https://www.aarda.org/diseaselist/). Table 2 lists the main indications of these medications, validated in both the USA and Europe, along with the restriction criteria of our research. All the English, Spanish, and French abstracts were reviewed by at least two senior researchers from our work group. American Society of Hematology educational books (http://asheducationbook.hematologylibrary.org/), textbooks of hematology and of internal medicine, and information gleaned from international meetings were also reviewed.

## 3. Definitions

Neutropenia is defined as an absolute blood neutrophil count ≤1.5 × 10^9^/L [2]. Severe neutropenia is defined as less ≤0.5 × 10^9^/L. In cases with neutropenia, patients are more susceptible to bacterial infections. Neutropenias have been classified in several categories based on the severity of neutropenia. One classification often used is based on neutropenia in oncology, with neutropenia categories from Grade 1 to 4 (‘Grade 1’: absolute neutrophil count from 1.5 to 1 × 10^9^/L; ‘Grade 2’: 1 to 0.5 × 10^9^/L; ‘Grade 3’: 0.5 to 0.1 × 10^9^/L; and ‘Grade 4’: ≤0.1 × 10^9^/L). Severe neutropenia, also called agranulocytosis, is characterized by a profound decrease or an absolute lack of circulating granulocytes, classically resulting in a neutrophil count of ≤0.5 × 10^9^/L (Table 3) [3]. For hematologists, a definition of ‘true’ agranulocytosis requires the combination of an absolute blood neutrophil count ≤0.1 × 10^9^ neutrophils per liter with fever or signs of sepsis [3]. To our knowledge, a majority of hospitalized patients for such neutropenia have a neutrophil count ≤0.1 × 10^9^/L, either initially or later in the hospitalization [2,3].

In practice, most but not all cases of neutropenia occur as a result of exposure to drugs, either chemotherapy (“chemotherapy neutropenia”) or other drugs (“idiosyncratic neutropenia”) [11,12]. Either the drug itself or one of its metabolites may be the causative agent [4]. These other drugs include all the commercialized biotherapies. In this setting, many causality assessment methods have been proposed to assess the relationship between a drug treatment and the occurrence of an adverse event, including neutropenia, in each patient [1,4]. The methods roughly belong to three categories: probabilistic approaches, algorithms, and expert judgment. For chemical drugs, the recommended criteria for blood cytopenias are derived from an international consensus meeting [3]. The criteria for assessing causality for implicating drugs in the etiology of neutropenia are outlined in Table 3 [1,13]. To date, these criteria remain valid for all chemical drugs. They are also applicable and valid for neutropenia related to biotherapy. However, there is at least one exception, that of late onset neutropenia with biotherapies (e.g., rituximab and alemtuzumab), which does not meet these criteria, but which is undoubtedly relevant to this subject of drug-induced neutropenia [14,15]. Late onset neutropenia is defined as a severe neutropenia (absolute neutrophil count ≤0.5 × 10^9^/L) occurring at least three or four weeks after biotherapy administration, mostly after a mean period of three months [14]. In the setting of drug-induced neutropenia, it should be emphasized that reintroducing the drug to prove the causality (theoretically the ‘gold standard’ for imputability criteria) is strictly proscribed.

## 4. Differential Diagnosis

In adults, the differential diagnosis of neutropenia with an absolute blood neutrophil count ≤1.5 × 10^9^/L includes a limited number of conditions [1,11,12]. Indeed, neutropenia with a neutrophil count ≤0.5 × 10^9^/L has been shown to be attributable to drugs in 70–90% of cases [11,12]. In the prospective Berlin Case-Control Surveillance Study on serious rare blood dyscrasias, idiosyncratic agranulocytosis was found to be drug-related in 97% of cases [11]. In clinical practice, the main differential diagnoses in adults, outside the context of chemotherapy and cancer, include: (i) neutropenia secondary to severe sepsis, particularly severe bacterial or viral infections; (ii) neutropenia manifesting as the first sign of bone marrow failure, such as in myelodysplastic syndromes (particularly in elderly patients) or acute leukemia; and (iii) neutropenia associated with hypersplenism (Table 4) [1,2,3]. Other, more rare differential diagnoses include: neutropenia secondary to peripheral destruction of polymorphonuclear cells, such as in Felty’s syndrome; large granular lymphocytic (LGL) leukemia; systemic lupus erythematosus and Sjögren’s syndrome (which is also often drug induced); or neutropenia secondary to nutritional deficiencies including cobalamin and vitamin B9 deficiencies.

For neutropenia related to biotherapy, this is important to keep in mind, particularly in autoimmune and auto-inflammatory diseases. In this setting, neutropenia may be initially related to several factors: the biological phenotype of several such diseases (e.g., seropositive and destructive rheumatoid arthritis, lupus erythematosus, Sjögren’s), particularly in chronic neutropenia; infections, often severe in frail patients or in cases of severe refractory heavily pre-treated disease, or in association with opportunistic infections; immunosuppressive agents (methotrexate, cyclophosphamide, azathioprine); and nutritional deficiencies (e.g., B9 vitamin in cases of methotrexate therapy) [14,16]. Therefore, the diagnosis of neutropenia related to biotherapy becomes a diagnosis of exclusion (Table 4).

## 5. Epidemiological Data

Idiosyncratic drug-induced neutropenia is a rare disorder. In Europe, the annual incidence of this hematological event is between 1.6 and 9.2 cases per million population [2,3]. In the USA, Strom et al. reported rates ranging from 2.4 to 15.4 per million per year [17]. Differences in the incidence may be due to different methods or inclusion criteria used in the studies published [1]. In our experience (observational study in a referral center), from 1996 to 2017, the annual incidence of drug-induced neutropenia remained stable, with around 6 to 7 cases per million population [3,18]. To our knowledge, the incidence remains unchanged, despite the introduction of new drugs (which carry a potential non-identified risk) and increased levels of medical awareness and vigilance (phase IV studies, systematic analyses of blood count when monitoring certain treatments). In general, most of the patients received more than two drugs (e.g., with a mean of three drugs in our cohort study), adding to the difficulty in definitively identifying the drug responsible for the neutropenia in clinical practice [1].

Outside the context of oncology, biotherapies have been reported as being the cause of early and transient Grade 1–2 neutropenia in 10–15% of the treated patients, often with TNF-α inhibitors [16,19,20,21,22,23,24], tocilizumab [25,26,27], rituximab [14,27,28,29,30,31,32,33,34,35,36], and alemtuzumab [15,37,38,39,40,41,42]. In a retrospective cohort study involving 499 patients with rheumatic diseases (rheumatoid arthritis in 72% of the diseases) treated by intravenous abatacept, infliximab, or tocilizumab, Espinoza et al. report at least one neutropenic episode in 52 patients (10.4%) [26]. Tocilizumab was more commonly associated with neutropenia than abatacept or infliximab (18.6% versus 3.8% and 2.8%, respectively, *p* < 0.001). In a recent study, Hastings et al. reported a 12.5–14.9% neutropenia rate among 367 patients under TNF-α inhibitors [16]. Rajakulendran et al. reported a 14.3% rate of idiosyncratic neutropenia in 133 patients with rheumatoid arthritis, without any other obvious cause other than anti-TNF-α treatment [24]. To our knowledge, the rate of TNF-α inhibitor-induced neutropenia has been comparable to that of neutropenia associated with commonly prescribed disease-modifying anti-rheumatic drugs (DMARD), such as methotrexate and leflunomide, with a neutropenia rate between 10 and 15% [16,19,20,21,22,23,24]. In autoimmune or auto-inflammatory diseases, idiosyncratic Grade 3–4 neutropenia and agranulocytosis are more rarely reported (1–2%), especially in rituximab therapy with late onset neutropenia and alemtuzumab [14,15]. Among 2624 rituximab-treated patients for refractory autoimmune and auto-inflammatory disorders and at least one follow-up visit, late onset neutropenia was observed in 40 patients (1.53%; 25 with rheumatoid arthritis (1.3% of these patients, 0.6/100 patient-years), and 15 with other rheumatologic disorders (2.3% of patients with these disorders, 1.5/100 patient-years)) [14]. Only a few case reports of Grade 3–4 neutropenia have been reported to date with anti-TNF-α therapy, tocilizumab therapy, and IL1 inhibitors [16,27,43]. To our knowledge, no severe neutropenia case has yet been reported with belimumab. For alemtuzumab, the European Medicines Agency reported neutropenia in 8.9% and 14.4% of multiple sclerosis (MS) patients (*n* = 811), after 1 year and 2 years, respectively [37]. The degree of neutropenia was generally mild, with only 0.6% of patients developing Grade 3–4 neutropenia at the 1-year follow-up and 1.5% after 2 years.

## 6. Drugs Involved

The drugs most commonly associated with idiosyncratic neutropenia are shown in Table 1 [1,2,3,44]. Almost all non-chemotherapy classes of drugs have been implicated, but for the majority the risk appears to be very small [2,3]. However, for drugs such as antithyroid medications, ticlopidine, clozapine, sulfasalazine, trimethoprim-sulfamethoxazole, and dipyrone, the risk may be higher [2,3]. For example, for antithyroid drugs, a risk of 3 per 10,000 users has been reported [45,46]. For ticlopidine, the risk is more than 100-fold higher [2,3]. Clozapine induces neutropenia in almost 1% of patients, particularly in the first three months of treatment, with older patients and females being at a higher risk [2,3]. In the context of hematology, deferiprone emerged as a causative agent of drug-induced neutropenia [2]. In our single center cohort (mentioned above), the most frequent causative chemical drugs, which are also frequently prescribed, were: antibiotics (49.3%), especially ß-lactams and cotrimoxazole; antithyroid drugs (16.7%); neuroleptic and anti-epileptic agents (11.8%); antiviral agents (7.9%); and platelet aggregation inhibitors such as ticlopidine (no longer used since the appearance of clopidogrel on the market) and acetylic salicylic acid (6.9%) (*n* = 203) [3]. These findings are similar to reports from several European research teams, except for antiviral agents [47,48,49]. In these studies, the most frequent causative types of chemical drugs were cotrimoxazole, carbimazole, ticlopidine, neuroleptic agents (clozapine), and nonsteroidal anti-inflammatory agents. It should be noted that the role of antibiotics or antiviral agents as causative drugs of neutropenia is often difficult to determine in the context of infection, especially in septicemia or severe sepsis and in viral infections (e.g., *Herpes viridae*, *Parvovirus B19 virus*) [3].

In autoimmune or auto-inflammatory disorders, several chemical drugs have been identified as the cause of neutropenia, including most of the nonsteroidal anti-inflammatory drugs and more rarely glucocorticoids; several DMARDs, particularly hydroxychloroquine, penicillamine, and sulfasalazine; and other drugs as colchicine and dapsone (Table 1) [1,2,3]. In this context, all biotherapies on the market in the USA and Europe have also been associated with neutropenia (especially well-documented for TNF-α inhibitors, tocilizumab, rituximab, and alemtuzumab) as described above (Table 1) [14,15,16,19,20,21,22,23,24,25,26,27,28,29,30,31,32,33,34,35,36,37,38,39,40,41,42,43]. For most of these biotherapies, the overall neutropenia risk is estimated to be around 10%, but this risk appears minimal (<1%) for Grade 3–4 neutropenia and agranulocytosis [2,14,15,16,19,20,21,22,23,24,25,26,27,28,29,30,31,32,33,34,35,36,37,38,39,40,41,42,43]. In this setting, the diagnosis of biotherapy-induced neutropenia may be difficult. The differential diagnosis of neutropenia in adults primarily concerns the underlying diseases (e.g., SLE, Felty’s syndrome) and their complications such as infections [2,14,16]. The impact of the chemical drugs, including nonsteroidal anti-inflammatory drugs, glucocorticoids, DMARDs, and immunosuppressive agents (e.g., methotrexate, cyclophosphamide), should be considered [14,16]. Concomitant infection may also play a role.

## 7. Risk Factors and Predisposing Conditions

As we have seen, much of the information about idiosyncratic drug-induced neutropenia comes from case reports, case-series, and epidemiological studies [1,2,3]. Inconsistent findings in terms of risk for a specific drug and predisposing conditions are usually found among different studies. The small size of such studies, particularly in the setting of autoimmune and auto-inflammatory diseases, makes predictions of the overall risk of this complication due to a specific drug very difficult [1]. For a few drugs, specific well-established risk factors for neutropenia have been identified, such as histocompatibility antigens (human leukocyte antigen [HLA]) [1,2]. For example, an association has been reported between HLA-B27 and HLA-B38 with the use of clozapine [49]. Conversely, the occurrence of HLA-B35 may prevent patients in certain ethnic groups from developing clozapine-induced agranulocytosis. Other risk factors include underlying autoimmune diseases such as rheumatoid arthritis in patients receiving captopril for renal failure, and concomitant treatment with probenecid [1,2].

The search for predisposing factors and conditions for this hematological event is important, particularly in an attempt to prevent or detect them early [1,3]. To date, few robust data are available with biotherapies [16]. In this setting, Hastings et al. performed a retrospective cohort study examining the association between baseline demographics, clinical features, medications used, development of neutropenia, and behavior of neutrophil counts upon TNF-α inhibitor therapy [16]. Their study included 367 patients given anti-TNF-α agents, mainly for RA (*n* = 298, 81.2%). Of these patients, 69 (18.8%) had at least one episode of neutropenia during TNF-α inhibitor treatment. In the study, patients with neutropenia exhibited significantly lower baseline blood neutrophil count levels (4.2 × 10^9^/L; 95% CI: 3.8, 4.6 × 10^9^/L), and a previous neutropenia history related to DMARD therapy increased the neutropenia risk upon receiving TNF-α inhibitors (hazard ratio 2.97; 95% CI: 1.69–5.25). The most significant predictor of developing neutropenia is a history of prior neutropenia when receiving a previous DMARD, given that these patients were three times more likely to develop neutropenia, especially during the first 3 months following treatment initiation [16,24]. In addition, a significant drop in mean blood neutrophil count was observed following 2 weeks of TNF-α inhibitor therapy, suggesting a bolus effect generated by the intravenous delivery of this drug. To our knowledge, no potential risk factors of neutropenia have been reported in patients treated with anti-CD20 agents outside the context of hematology [50]. Nevertheless, among the baseline characteristics in rheumatoid arthritis patients (*n* = 2624), only female sex and age have been associated with rituximab-induced neutropenia [14]. No other studied factor, including disease duration, rheumatoid factor, anti-cyclic citrullinated peptide activity, numbers of previous synthetic DMARDs, and anti-TNF disease activities at enrollment, concomitant treatment with DMARD or corticosteroids, and serum gamma globulin and immunoglobulin G (IgG) levels, has been associated. In this setting, the following factors were identified as neutropenia predictors with intravenous DMARD: a history of neutropenia with methotrexate (OR 1.56, 95% CI: 1.17–7.14), concomitant methotrexate treatment (OR 1.21, 95% CI: 1.01–2.64), and tocilizumab treatment (OR 2.72, 95% CI: 1.53–9.05) [36].

## 8. Pathogenesis

Little work has been devoted in recent years to the study of the pathophysiology of drug-induced neutropenia [4,5,6,7,8,9,10,11,12,13,14,15,16,17,18,19,20,21,22,23,24,25,26,27,28,29,30,31,32,33,34,35,36,37,38,39,40,41,42,43,44,45,46,47,48,49,50,51,52,53]. Clinical observations, studies in volunteers, and laboratory experiments have suggested that idiosyncratic drug-induced neutropenia is mediated by immune allergic and toxic mechanisms [4,5,6,7,8,9,10,11,12,13,14,15,16,17,18,19,20,21,22,23,24,25,26,27,28,29,30,31,32,33,34,35,36,37,38,39,40,41,42,43,44,45,46,47,48,49,50,51,52,53]. This is also the case for the biotherapy family of drugs [16,24,25,29,36]. The pathogenesis is however a heterogeneous process that is not yet fully understood. In many cases, neutropenia occurs after prolonged drug exposure, resulting in decreased granulocyte production by a hypoplastic bone marrow [4]. In other cases reported, intermittent exposure is implicated. This suggests an immune mediated mechanism (involving cytotoxic T-cells, haptens, or autoimmunity), although this hypothesis is not entirely confirmed. Direct damage, either to the microenvironment of the bone marrow or to myeloid precursors, may also play a significant role in most other cases [4,5,6,7,8,9,10,11,12,13,14,15,16,17,18,19,20,21,22,23,24,25,26,27,28,29,30,31,32,33,34,35,36,37,38,39,40,41,42,43,44,45,46,47,48,49,50,51,52,53]. Genetic polymorphism has been considered, given the heterogeneity of expression of the various enzymes that metabolize drugs and other chemicals, as well as oxidative modification of the drug [4,5,6,7,8,9,10,11,12,13,14,15,16,17,18,19,20,21,22,23,24,25,26,27,28,29,30,31,32,33,34,35,36,37,38,39,40,41,42,43,44,45,46,47,48,49,50,51,52,53]. The impact of myeloperoxidase and NADPH-oxidase polymorphism in drug-induced agranulocytosis has been studied [4,5,6,7,8,9,10,11,12,13,14,15,16,17,18,19,20,21,22,23,24,25,26,27,28,29,30,31,32,33,34,35,36,37,38,39,40,41,42,43,44,45,46,47,48,49,50,51,52,53].

The mechanisms of neutropenia related to biotherapies are not yet fully elucidated. However, one hypothetic mechanism for anti-TNF-α therapy-induced neutropenia relies on the impact of drugs that serve as haptens and sensitize neutrophils or neutrophil precursors, resulting in immune-mediated peripheral destruction [16,24,25]. In addition, induced circulating anti-neutrophil antibodies are likely to cause increased peripheral destruction, which is often seen in some viral infections [16,24,25]. The pathophysiology of rituximab-induced neutropenia also remains unclear, given that neutrophils do not express CD20 [29]. Several assumptions have been made concerning the pathogenesis of neutropenia following rituximab administered to B-cell lymphoma patients, including the role of antinuclear antibodies, large granular lymphocytes, competition for growth factors between lymphopoiesis and granulopoiesis, as well as genetic polymorphisms in the IgG receptor FCƔ RIIIA [14,29]. The upregulation of the TNF family B-cell activating factor following B-cell depletion may similarly play a role by favoring B-cell repopulation to the detriment of granulopoiesis [35]. For tocilizumab, several different mechanisms underlying an absolute neutrophil count decrease have been proposed, including bone marrow suppression, accelerated peripheral apoptosis, and intravascular neutrophil margination [25,26]. However, the rapid neutropenia onset that sometimes manifests within several hours following tocilizumab administration does not suggest bone marrow involvement. Another possible explanation of tocilizumab-induced neutropenia is that tocilizumab may increase neutrophil apoptosis, subsequently decreasing the circulating pool of neutrophils. Alemtuzumab-related neutropenia has been shown to require a long time interval from the initial drug treatment and may occur due to secondary autoimmunity [27,36]. The risk of developing secondary autoimmunity is greatest within the first five years of follow-up. Thus, neutropenia is typically delayed and occurs after immune reconstitution [8]. The immune-mediated mechanism related to this drug is further confirmed by the responsiveness to corticosteroids.

## 9. Clinical Manifestations

Patients with drug-induced neutropenia usually are asymptomatic or present with fever (often the earliest sign), associated with general malaise (often including chills, myalgia, and/or arthralgia) with a non-specific sore throat, and other localized infections [1,2,3,54]. For Grade 3–4 neutropenia and agranulocytosis, most patients (>60%) who do not receive medical intervention develop septicemia, while some have clinical signs of pneumonia as well as anorectal, skin, or oropharyngeal infections and septic shock [1,3,54]. In our aforementioned cohort study (*n* = 203), the main clinical presentations during hospitalization were: isolated fever or fever of unknown origin (26.3%), septicemia (13.9%), documented pneumonia (13.4%), sore throat and acute tonsillitis (9.3%), and septic shock (6.7%) [3]. The remaining symptomatic patients presented documented infections (21.7%), including cutaneous infections, deep abdominal or thoracic abscess, and acute pyelonephritis. While in hospital, 19.2% of the patients worsened clinically and exhibited features of severe sepsis, septic shock, or systemic inflammatory response syndrome (SIRS) [3]. As in patients receiving chemotherapy for the treatment of cancer, the occurrence of infections depends on the degree and duration of the neutropenia and the patient phenotype (age, medical history, and type of comorbidities) [1,55,56]. It is notable that when antibiotics are administered prophylactically, both the patient’s complaints and the physical findings may be “masked” and fever may be the only clinical sign detected [2,3]. It should be noted that the follow-up of the patients potentially modifies the mode of discovery of the neutropenia, with asymptomatic patients or patients with isolated fever, but in our experience without modifying the evolution of this hematological event [1,3]. In elderly patients, clinical manifestations are generally more severe, with septicemia or septic shock in at least four-fifths of patients, and the outcomes are worse [57].

Unlike other drugs, only a minority of patients treated with biotherapies develop a neutropenia that is symptomatic or results in a sepsis [14,15,16,19,20,21,22,23,24,25,26,27,28,29,30,31,32,33,34,35,36,37,38,39,40,41,42]. To our knowledge, the mild nature of the infection (fever, painful swallowing, gingival pain, skin abscesses, sore throats, and otitis) has not yet been explained. It is true that a majority of neutropenias reported with biotherapies are Grade 1–2 and are diagnosed early. However, most of the patients concerned are heavily pretreated and have severe refractory autoimmune or auto-inflammatory disorders. Exceptional cases of severe infections have nevertheless been reported, especially with rituximab, but also with anti-TNF-α therapy, and more rarely with alemtuzumab and IL-1 inhibitors [14,19,20,21,22,23,43]. For rituximab, 12.5–66.7% of patients with Grade 3–4 neutropenia related to anti-CD20 agents, especially late onset neutropenia related to rituximab, develop severe, even life-threatening, complications, including pneumonias, septicemia, and invasive infections [14,29,30,31,32,33,34]. In the study by Salmon et al., five patients (12.5% of the neutropenic patients) developed a serious non-opportunistic infection and required antibiotics and Granulocyte-Colony Stimulating Factor (G-CSF) injections, with a favorable outcome [14]. Ogawa et al. reported four cases (36.4%) of late onset neutropenia following rituximab therapy [32]. Three cases required G-CSF, whereas no severe infections developed. Ahmadi et al. reported late onset neutropenia in four of the six rituximab-treated kidney transplant recipients with antibody-mediated rejection (66.7%) [33]. The course of neutropenia was complicated by endocarditis in one patient, resulting in his death due to the lack of valvular surgery. We also documented one such case of late onset neutropenia (blood neutrophil count ≤0.1 × 10^9^/L) in a 58-year-old patient treated with corticosteroids and rituximab for refractory Evans syndrome [58]. In this patient, an invasive aspergillosis (*Aspergillus fumigatus*) of the hand occurred and was cured using triazole antifungal agents. The late onset neutropenia required GSF treatment in addition to rituximab cessation. To date, this is not currently the case for another anti-CD20 antibody called obinutuzumab or other rituximab biosimilars (e.g., CPT-10) [59]. In the Hasting et al. study (*n* = 298), only 6% of the studied patients later developed serious infections secondary to neutropenia induced by TNF-α inhibitors [16]. In this setting, Guiddir et al. reported a case series involving four newborn patients with severe neutropenia born to infliximab-treated mothers for ulcerative colitis during pregnancy, including the third trimester [22]. The newborns presented with Grade 3–4 neutropenia at birth that was subsequently complicated by skin infections. Yiannopoulou et al. reported the first case of alemtuzumab-infusion-related death due to early neutropenia in a non-immunocompromised multiple sclerosis (MS) patient [15]. A 47-year-old Caucasian female received alemtuzumab after a serious relapse of her relapsing-remitting MS. At 23 days after alemtuzumab infusion, she developed severe early agranulocytosis, which resulted in septic shock by *Staphylococcus aureus* and death. To our knowledge, only one case report of agranulocytosis under tocilizumab therapy has been reported to date [27]. The case concerned a rheumatoid patient in whom agranulocytosis manifested after the 74th tocilizumab course in the context of a parvovirus B19 infection.

## 10. Prognosis and Mortality Rate

Over the past 20 years, the mortality rate for idiosyncratic drug-induced neutropenia was 10–16% in European studies [1,2,3]. This is likely due to improved recognition, management, and treatment of the condition [1,2,3]. Grade 3–4 neutropenias are the most likely to cause death, as in oncology, where the severity of neutropenia has a documented impact on prognosis. To date, no robust data are available in the context of autoimmune or auto-inflammatory diseases. This is due to the low number of documented cases available and the small number of patients in series. To our knowledge, only two deaths have been reported with neutropenia secondary to the use of rituximab and alemtuzumab biotherapy [33,39].

With chemical drugs, the highest mortality rate is observed in older patients (≥65 years), as well as in those with renal failure (defined as serum creatinine level ≥120 µmol/L), bacteremia, or shock at the time of diagnosis [2,56]. Table 5 presents factors influencing the prognosis (hematological recovery, duration of hospitalization and antibiotic therapy, and mortality) [3,56]. We have previously confirmed these findings by performing a uni- and multivariate analysis of factors affecting the outcome in our cohort study (*n* = 91) [56]. Specifically, we found that a blood neutrophil count of ≤0.1 × 10^9^/L at the time of diagnosis, as well as septicemia and/or shock, were variables that were significantly associated with a longer neutrophil recovery time. In contrast, the use of hematopoietic growth factors was associated with a shorter neutrophil recovery time [3]. We have not significantly documented the impact of underlying disease. In the systematic review by Andersohn et al. [2], which included 492 published case reports of agranulocytosis, it was shown that patients with a blood neutrophil count of ≤0.1 × 10^9^/L had a higher rate of localized infections (59% versus 39%, *p* < 0.001), sepsis (20% versus 6%, *p* < 0.001), and fatal complications (10% versus 3%, *p* < 0.001) than those with a neutrophil nadir ≥0.1 × 10^9^/L. Julia et al. have shown previously that the result of a bone marrow analysis is a predictor of the neutrophil recovery [54]. Thus, in cases where there is a lack of myeloid precursors, blood count recovery is unlikely to occur before 14 days, whereas in cases of maturation arrest, recovery generally occurs within 2 to 7 days. To date, no identified risk factor for admission to the intensive care unit and/or death was determined with chemical drugs or with biotherapies. Nevertheless, the impact of Grade 3–4 neutropenia and the existence of auto-immune or auto-inflammatory disorders must not be zero, even if not documented to date with an appropriate methodology, in patients who are often fragile with severe underlying diseases or even refractory to conventional therapies.

## 11. Management

The management of drug-induced neutropenia begins with the immediate withdrawal of any medications which may potentially be responsible [1,2,3]. The patient’s medication history must be carefully obtained in chronological order so that the suspected agent(s) may be identified. Importantly, the appropriate pharmacovigilance center must be notified of all cases of drug-induced neutropenia [1].

In the setting of biotherapy, only a minority of induced neutropenia cases are considered severe. These later usually appear to be transient and self-limited [14,15,16,19,20,21,22,23,24,25,26,27,28,29,30,31,32,33,34,35,36,37,38,39,40,41,42,43]. The management of neutropenic episodes caused by biotherapy consists of the following: (i) continuation of the original biotherapy in cases of Grade 1 neutropenia with strict monitoring; (ii) temporary cessation of the original drug and reinstatement once neutrophil count has returned to normal level for Grade 2 neutropenia; and (iii) switching to an alternative agent, while definitively stopping the biotherapy, in cases of Grade 3–4 neutropenia or severe sepsis [2,14,15,16,19,20,21,22,23,24,25,26,27,28,29,30,31,32,33,34,35,36,37,38,39,40,41,42,43]. The transient and moderate depth of neutropenia in most cases might explain why most of the patients with a prior biotherapy-induced neutropenia may be able to be re-treated with the same biotherapeutic agent. In the aforementioned study from Rajakulendran et al., 84.2% of the 133 patients continued to receive their initial anti-TNF-α therapy, with only one temporary cessation [24]. In this study, one infliximab-receiving patient had recurrent episodes of neutropenia that were managed by means of temporary cessation. Subsequently, however, the patient was switched to etanercept, with no further neutropenia episodes since. Another patient was switched from etanercept to adalimumab, without further problems. In the Hasting et al. study, no new neutropenia episodes were described once patients were switched to another anti-TNF-α [16]. For anti-CD20 agents, a recurrent episode of uncomplicated neutropenia was observed in only 16% of re-treated patients in the Salmon et al. study, with only one single patient exhibiting a Grade 3 neutropenia [14]. Overall, 19 patients (47.5%) with previous rituximab-induced neutropenia received a new rituximab infusion following resolution of their neutropenia. While Grade 1–2 neutropenia reoccurred in three patients, none of these neutropenia recurrences required G-CSF or was complicated by an infection.

In the setting of drug-induced neutropenia (chemical drugs or biotherapies), the occurrence of sepsis requires prompt management, including administration of antibiotics and hospitalization [1,2,3]. It should be noted that, as a result of neutrophil deficiency, both the patient’s symptoms and the physical findings may be altered, and fever may be the only clinical sign. Asymptomatic patients at high risk of infection should also be admitted to the hospital [1,3]. Important prognostic factors resulting in an increased risk of serious complications are displayed in Table 5 for all chemical drugs and above in the text [54,56].

Even patients with a low risk of infection, with none of these risk factors and good general health, should be treated in the hospital, unless adequate and comprehensive medical follow-up can be provided in an ambulatory setting or at home [1]. Preventive measures include good hygiene and infection control, paying attention to high-risk areas such as the mouth, skin, and perineum [1,2,3]. Patient isolation and the use of prophylactic antibiotics (e.g., for the gastrointestinal tract) have been proposed, but their usefulness in limiting the risk of infection has not been clinically proven [1]. Concomitant measures include aggressive treatment of confirmed or potential sepsis, as well as the prevention of secondary infections [1,2,3,54].

The impact of immune impairment in autoimmune and auto-inflammatory disorders on infection risk is not fully understood. At present, the only recommended preventive measures consist of hepatitis B and C vaccination, a Listeria-free diet, tuberculosis screening and prophylaxis, annual papillomavirus screening for all biotherapies, and anti-herpetic prophylaxis for alemtuzumab [14,15,16,37]. Given the non-negligible risk of unpredicted infective events, Buonomo et al. advised physicians to take into account patients’ history of infectious diseases and vaccine status and to consider supplementary prophylactic strategies, including screening for *Toxoplasma gondii* and viral hepatitis serological status, as well as pre-emptive approaches to avert both CMV reactivation and pneumocystosis [60].

The occurrence of sepsis requires prompt management, including the administration of broad-spectrum intravenous antibiotic therapy (after blood, urine, and any other relevant samples have been cultured [1,2,3,54]). Empiric, broad-spectrum antibacterial therapy is generally the best choice, but the choice of antibiotic used may need to be adapted depending on the nature of the sepsis, the clinical status of the patient, local patterns of antibiotic resistance, and previous antibiotic use [1,2,3]. In the setting of biotherapy-induced neutropenia, this is even more important as patients are often fragile, with more severe forms of the disease, and heavily pre-treated [14,15,16]. When an antibiotic is suspected of being the causative agent resulting in immune mediated neutropenia, one should keep in mind the potential for antibody cross-reactivity, and therefore the choice of further antibiotics to be administered should be considered very carefully [3].

In drug-induced neutropenia, the successful use of the hematopoietic growth factors (HGF), particularly the G-CSF, has been previously reported [1,2,61]. Since 1985, two-thirds of reported cases have been treated with HGF [61]. The most recent major studies on HGF use in drug-induced agranulocytosis are described in Table 6 [1,2,3]. G-CSF (at a mean dose of 5 µg/kg/day) was found to be useful in shortening the duration of blood count recovery time, without inducing any major toxic or adverse effects, particularly in patients with poor prognostic factors [1,3,56,61]. We have also demonstrated that the duration of antibiotic therapy and hospital stay are significantly shorter in patients treated with HGF [1,3]. The systematic review of all published case reports of non-chemotherapy drug-induced agranulocytosis by Andersohn et al. (*n* = 492) confirms this data [2]. The study by Ibanez et al. (Barcelona cohort) also concludes that G-CSF shortens recovery time in patients with agranulocytosis [44]. In this setting, only the study of Beauchesne et al. reported a lower mortality rate with this therapy [62]. Of note, the only prospective randomized study available did not confirm the benefit of G-CSF [63]. In an update of the aforementioned cohort study, we have documented that the mean duration of hematological recovery was reduced to 2.1 days (range: 2–16) in patients treated with HGF (*n* = 107) (*p* = 0.057) [3]. The mean duration of antibiotherapy and hospitalization are not improved by the use of HGF: 22.3 (range: 7–120) and 30.9 (range: 5–200) days, respectively (all *p* > 0.4).

As we have seen above, only a minority of biotherapy-induced neutropenia cases are considered severe (Grade 3–4), including late-onset neutropenia [14,15]. In this setting, several authors have reported successful treatment using G-CSF, while others have observed blood cell count recovery after discontinuation of treatment [14,15,16,19,20,21,22,23,24,25,26,27,28,29,30,31,32,33,34,35,36,37,38,39,40,41,42,43]. For example, this is the case for alemtuzumab-related Grade 3–4 neutropenia in patients with kidney transplantation. In this setting, most of the patients (61.5%) required G-CSF for recovery [33].

In chemical drug-induced agranulocytosis, therapeutic measures, such as transfusions of granulocyte concentrates, should only be used in exceptional circumstances, and only then for the control of life-threatening infections with antibiotic resistance such as perineal gangrene [1,2,3].

## 12. Prevention

Routine monitoring of neutrophil count in the general population is not indicated [1,2,54]. However, routine monitoring for neutropenia is at least recommended, and perhaps strictly required, in the use of some high-risk drugs such as clozapine, ticlopidine, and antithyroid drugs [2,64,65]. When prescribing antithyroid agents, a standardized approach with neutrophil count examination at each visit was recently shown to correctly diagnose 64% and 94% of patients with agranulocytosis with no or minimum infection symptoms, respectively [65]. To date, this recommendation continues to be debated because of the absence of impact on mortality and morbidity [3]. This may explain current attitudes towards routine monitoring of blood counts even in individuals receiving high risk medications such as antithyroid drugs or ticlopidine [1,2].

In the case of biotherapy, a routine monitoring of the neutrophil count is generally performed in the evaluation of the underlying disease, often severe and heavily pre-treated, in patients who are often very fragile. Thus, routine monitoring is recommended for all biotherapy products.

## 13. Conclusions

Today, drug-induced neutropenia remains a potentially serious adverse event due to the frequency of severe sepsis, with severe deep tissue infections (e.g., pneumonia), septicemia, and septic shock in at least two-thirds of all hospitalized patients, particularly those with Grade 3–4 neutropenia. Knowledge of the commonly-implicated agents and a high index of suspicion are essential in diagnosis. This is particularly important in the setting of autoimmune and auto-inflammatory disorders where the causes of neutropenia can be multiple. In this setting, transitory Grade 1–2 neutropenia related to biotherapy is relatively common with several biotherapies (e.g., TNF-alpha inhibitors, IL6 inhibitors, and anti-CD52 agents). With biotherapies, Grade 3–4 neutropenia or agranulocytosis and clinical manifestations related to sepsis are the exception, with to date only a few published case reports. Special mention should be made of late-onset and potentially severe neutropenia, especially following anti-CD52 agent therapy. Physicians must be vigilant in identifying drug-induced neutropenia because early detection can decrease the severity and prevent mortality if the drug is discontinued. Over the past 10 years, several prognostic factors have been identified which may be helpful for the management of patients with ‘suspected’ or ‘proven’ agranulocytosis. Older age, septicemia or shock, metabolic disorders such as renal failure, and a neutrophil count ≤0.1 × 109/L have been accepted as poor prognostic factors.

Currently, the number of drugs that adversely affect the blood system continues to increase and their effects, especially in cases of severe neutropenia, pose a great challenge to all physicians. This effect is well known for several drugs, including antithyroid drugs, ticlopidine, clozapine, sulfasalazine, antibiotics including trimethoprim-sulfamethoxazole, and deferiprone, but new agents have been implicated as well. Moreover, given the advancing age of the population, the increasing use of medications as a therapeutic modality, and the subsequent longer exposure to drugs, as well as the development of new agents, health care professionals should be aware of this adverse event and its management.

## Figures and Tables

**Table 1 jcm-08-01351-t001:** Drugs related to idiosyncratic neutropenia and agranulocytosis [1,2,3].

Drug Family	Drugs
Analgesics and non-steroidal anti-inflammatory drugs	Acetaminophen, acetylsalicylic acid (aspirin), aminopyrine, benoxaprofen, diclofenac, diflunisal, dipyrone, fenoprofen, indomethacin, ibuprofen, naproxen, phenylbutazone, piroxicam, sulindac, tenoxicam, tolmetin
Antipsychotics, hypnosedatives and antidepressants	Amoxapine, chlomipramine, chlorpromazine, chlordiazepoxide, clozapine, diazepam, fluoxetine, haloperidol, levomepromazine, imipramine, indalpine, meprobamate, mianserin, olanzapine, phenothiazines, risperidone, tiapride, ziprasidone
Antiepileptic drugs	Carbamazepine, ethosuximide, phenytoin, trimethadione, valproic acid (sodium valproate)
Antithyroid drugs	Carbimazole, methimazole, potassium perchlorate, potassium thiocyanate, propylthiouracil
Cardiovascular drugs	Acetylsalicylic acid (aspirin), amiodarone, aprindine, bepridil, captopril, coumarins, dipyridamole, digoxin, flurbiprofen, furosemide, hydralazine, lisinopril, methyldopa, nifedipine, phenindione, procainamide, propafenone, propranolol, quinidine, ramipril, spironolactone, thiazide diuretics, ticlopidine, vesnarinone
Anti-infective agents	Abacavir, acyclovir, amodiaquine, atovaquone, cephalosporins, chloramphenicol, chloroguanine, chloroquine, ciprofloxacin, clindamycin, dapsone, ethambutol, flucytosine, fusidic acid, gentamicin, hydroxychloroquine, isoniazid, levamisole, lincomycin, linezolid, macrolides, mebendazole, mepacrine, metronidazole, minocycline, nitrofurantoin, norfloxacin, novobiocin, penicillins, pyrimethamine, quinine, rifampicin, streptomycin, terbinafine, tetracycline, thioacetazone, tinidazole, trimethoprim-sulfamethoxazole (cotrimoxazole), vancomycin, zidovudine
Biotherapies	Anti-CD20 agents (rituximab), anti-CD52 (alemtuzumab), interleukin-1 inhibitors (anakinra, canakinumab), interleukine-6 inhibitors (tocizulimab), interferon-α, TNF-α inhibitors (adalimumab, etanercept infliximab)
Miscellaneous drugs	Acetazolamide, acetylcysteine, allopurinol, aminoglutethimide, arsenic compounds, bezafibrate, brompheniramine, calcium dobesilate, chlorpheniramine, cimetidine, colchicine, dapsone, deferiprone, famotidine, flutamide, gold, glucocorticoids, hydroxychloroquine, mesalazine, methapyrilene, methazolamide, metoclopramide, levodopa, octreotide, olanzapine, omeprazole, oral hypoglycemic agents (glibenclamide), mercurial diuretics, penicillamine, ranitidine, riluzole, sulfasalazine, most sulfonamides, tamoxifen, thenalidine, tretinoin, tripelennamine

**Table 2 jcm-08-01351-t002:** Autoimmune and auto-inflammatory diseases, systemic vasculitis, and orphan diseases using biotherapies for their treatment (https://www.aarda.org/diseaselist/).

Adult Still’s disease
Amyloidosis
Ankylosing spondylitis
Antiphospholipid syndrome
Behçet’s disease
Churg–Strauss Syndrome (CSS) or Eosinophilic Granulomatosis (EGPA)
Cold agglutinin disease
CREST syndrome
Crohn’s disease
Dermatomyositis
Evans syndrome
Giant cell arteritis (temporal arteritis)
Granulomatosis with polyangiitis (Wegener’s granulomatosis)
Hemolytic autoimmune anemia
IgG4-related sclerosing disease or hyper-IgG4 syndrome
Immune thrombocytopenic purpura
Juvenile arthritis
Kawasaki disease
Lupus
Microscopic polyangiitis (MPA)
Mixed connective tissue disease (MCTD)
Multiple sclerosis
Myasthenia gravis
Neutropenia (autoimmune)
Paroxysmal nocturnal hemoglobinuria (PNH)
Polyarteritis nodosa
Polymyalgia rheumatica
Polymyositis
Pure red cell aplasia (PRCA)
Relapsing polychondritis
Rheumatoid arthritis (RA)
Sarcoidosis
Scleroderma
Sjögren’s syndrome
Takayasu’s arteritis
Temporal arteritis/giant cell arteritis
Thrombocytopenic purpura (TTP)
Ulcerative colitis (UC)
Undifferentiated connective tissue disease (UCTD)
Uveitis
Vasculitis

**Table 3 jcm-08-01351-t003:** Definition and criteria of drug imputability for idiosyncratic chemical drug-induced neutropenia and agranulocytosis (adapted from [1,2]).

Definition of Neutropenia and Agranulocytosis	Criteria of Drug Imputability
▪ Neutropenia is defined by a neutrophil count ≤1.5 × 10^9^/L ▪ Agranulocytosis is defined by a neutrophil count ≤0.5 × 10^9^/L ± existence of a fever and/or any signs of infection	▪ Onset of agranulocytosis during treatment or within 7 days after exposure to the drug, with a complete recovery in neutrophil count of more than 1.5 × 10^9^/L within one month of discontinuing the drug ▪ Recurrence of agranulocytosis upon re-exposure to the drug (theoretically the gold method but ethically questionable) ▪ Exclusion criteria: history of congenital neutropenia or immune mediated neutropenia, recent infectious disease (particularly recent viral infection), recent chemotherapy and/or radiotherapy and/or biotherapy * and existence of an underlying hematological disease

*: Intravenous polyvalent immunoglobulins, interferon, anti-TNF, anti-CD20 (rituximab).

**Table 4 jcm-08-01351-t004:** Differential diagnosis of biotherapy-induced neutropenia in adults [1,14,16].

− Normal variations: Ethnic and familial neutropenia
− Splenic sequestration: Cirrhosis and portal hypertension (alcoholism), Gaucher’s disease
− Nutritional deficiencies: Cobalamin and folate deficiencies, copper deficiency, cachexia (Kwashiorkor)
− Infections: Bacterial (typhoid fever, brucellosis, tuberculosis, rickettsia, severe sepsis), viral (Epstein–Barr virus, cytomegalovirus, human immunodeficiency virus, hepatitis virus, rubella, parvovirus B19), protozoal and fungal (histoplasmosis, leishmaniasis, malaria)
− Other drugs intake: especially ticlopidine, clozapine, sulfasalazine, trimethoprim-sulfamethoxazole (cotrimoxazole), and dipyrone (target the last introduced drug)
− Immune neutropenia: Isolated autoimmune neutropenia, collagen vascular autoimmune disease (systemic lupus erythematosus, rheumatoid arthritis, or Felty’s syndrome), T γ-δ lymphocytosis
− Hematological disease: Myelodysplasia, pure white blood cell aplasia and red cell aplasia, Marchiafava–Michelli disease
− Primary congenital or chronic neutropenia: Familial and nonfamilial cyclic neutropenia

**Table 5 jcm-08-01351-t005:** Impact factors for the prognosis * of idiosyncratic drug-induced agranulocytosis (adapted from [1,3,56]).

▪ Age: >65 years	Negative impact on duration of hematological recovery **, duration of hospitalization and antibiotherapy
▪ Neutrophil count at diagnosis: ≤0.1 × 10^9^/L	Negative impact on duration of hematological recovery, duration of hospitalization and antibiotherapy
▪ Clinical status: Deep severe infections or bacteremia or septic shock (versus isolated fever)	Negative impact on duration of hospitalization and antibiotherapy and of mortality
▪ Severe underlying disease or severe co-morbidity: Renal failure, cardiac or respiratory failure, systemic auto-inflammatory diseases	Negative impact on duration of hematological recovery and hospitalization
▪ Management with pre-established procedures and hematopoietic growth factor for use in severe conditions	Positive impact on duration of hematological recovery, duration of hospitalization and of mortality

* Prognosis: hematological recovery, duration of hospitalization and antibiotherapy, mortality. ** Hematological recovery: absolute neutrophil count >1.5 × 10^9^/L.

**Table 6 jcm-08-01351-t006:** Recent studies on the use of hematopoietic growth factors in idiosyncratic chemical drug-induced agranulocytosis [1,2,3,61,63].

Type of Study and Target Population	Main Results
Systematic review of all published cases (*n* = 492); All patients with idiosyncratic drug-induced agranulocytosis (Andersohn F. et al. Ann. Intern. Med. 2007, 146, 657–665)	Treatment with hematopoietic growth factors was associated with a statistically significantly lower rate of infectious and fatal complications, in cases with a neutrophil count <0.1 × 10^9^/L.
Meta-analysis (*n* = 118); All patients with idiosyncratic drug-induced agranulocytosis (Ibáñez L. et al. Drug Saf. 2008, 17, 108–109)	G-CSF or GM-CSF (100 to 600 µg/day) reduced the mean time to neutrophil recovery (neutrophil count >0.5 × 10^9^/L) from 10 to 7.7 days, in cases with a neutrophil count <0.1 × 10^9^/L, and reduced the mortality rate from 16 to 4.2%.
Case control study, retrospective analysis (*n* = 70); All patients with idiosyncratic drug-induced agranulocytosis (Sprikkelman A. et al. Leukemia 1994, 8, 2031–2036).	G-CSF and GM-CSF (100 to 600 µg/day) reduced the recovery of neutrophil count from 7 to 4 days, particularly in patients with a neutrophil count <0.1 × 10^9^/L.
Cohort study, retrospective analysis (*n* = 54); Patients with idiosyncratic drug-induced agranulocytosis >65 years of age, with poor prognostic factors (Andrès E. et al. Am. J. Med. 2002, 112, 460–464)	G-CSF (300 µg/day) significantly reduced the mean duration for hematological recovery from 8.8 to 6.6 days (*p* < 0.04). G-CSF reduced the global cost.
Cohort study, retrospective analysis (*n* = 20); Patients with antithyroid drug-induced agranulocytosis and poor prognostic factors (Andrès E. et al. QJM 2001, 94, 423–428)	G-CSF (300 µg/day) significantly reduced the mean durations of hematological recovery, antibiotic therapy and hospitalization from: 11.6 to 6.8 days, 12 to 7.5 days and 13 to 7.3 days, respectively (*p* < 0.05 in all cases). G-CSF reduced the global cost.
Cohort study, retrospective analysis (*n* = 145); All patients with idiosyncratic drug-induced agranulocytosis (Ibáñez L. et al. Pharmacoepidemiol. Drug Saf. 2008, 17, 108–109)	G-CSF shortens time to recovery in patients with agranulocytosis.
Cohort study, retrospective analysis (*n* = 201); All patients with idiosyncratic drug-induced agranulocytosis (Andrès E. et al. QJM 2017, 110, 299–305)	G-CSF (300 µg/day) reduced the mean durations of hematological recovery for 2.1 days (*p* = 0.057).
Prospective randomized study (*n* = 24); All patients with antithyroid drug-induced agranulocytosis (Fukata S. et al. Thyroid 1999, 9, 29–31)	G-CSF (100 to 200 µg/day) did not significantly reduce the mean duration for hematological recovery.

G-CSF: Granulocyte-colony stimulating factor. GM-CSF: Granulocyte-macrophage-colony stimulating factor.

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
