# Peer review of "State of Art of Idiosyncratic Drug-Induced Neutropenia or Agranulocytosis, with a Focus on Biotherapies"

_jcm, 2019, doi:10.3390/jcm8091351_

Round 1
Reviewer 1 Report
This is a well written, comprehensive review of idiosyncratic drug-induced neutropenia and agranulocytosis, beginning with a thorough differentiation between neutropenia of any Grade and agranulocytosis. There is a focus on the more recently developed biotherapy drugs.
1. Remove “of,” line 97.
2. The focus is on adult patients affected with drug-induced neutropenia. Do the authors care to mention what is known about this disorder in children?
Author Response
Thank you so much for your comments:
1st point: of removed from line 97
2nd point: We have no data about this disorders in children
Reviewer 2 Report
The authors review the current data about biotherapy outside the context of oncology and its risk to induce neutropenia. The paper gives a broad overview over biotherapies and other drugs with known risk for neutropenia. While drugs such as antithyroid drugs, clozapine, dipyrone, and several antibiotics have been discussed frequently in the past, respective information about biotherapy used for autoimmune and anti-inflammatory therapy is currently lacking. The current paper therefore focuses on this area and gives interesting and important information about the risk for neutropenia related to biotherapy as well as the treatment options and recommendations for treating physicians. The authors highlight in detail the difficulties of differential diagnosis regarding biotherapy related neutropenia.
Broad comments
The epidemiologic data (chapter 4) and the clinical manifestations (chapter 8) would benefit from an additional table, which summarizes the described data. This would make the data clearer and the text easier to understand.
Is there information about co-medication with methotrexate in addition to the biotherapy for the cases mentioned in the cited studies?
Can the authors make an assumption, why late onset neutropenia (e.g. after rituximab treatment) is often more severe?
Specific comments
Table 3: do you mean immune mediated neutropenia or auto-immune mediated neutropenia as exclusion criterion?
Page 8, lines 232-238: There is a recent study, that shows that common risk factors, such as autoimmune diseases and comedication, may have no impact on neutropenia development for the analgesic dipyrone (doi: 10.1016/j.ejim.2019.07.029 ). It would be interesting to also mention this possibility of drug induced neutropenia.
Page 9, lines 269-278: Please give more than one reference for this text passage about pathogenesis.
Page 9, line 279: remove “the” before neutropenia
Page 9, lines 280-282: the mentioned references do not show the effect of haptens, the mechanism is unknown. Instead, other possible mechanisms could be discussed as reported by Shovman et al. 2015 (Ref. 25).
Page 9, lines 282-284: Please give a reference for this information.
Page 13, line 455: does antibody cross reactivity of antibiotics need to be considered for every antibiotics induced neutropenia or only for immune mediated neutropenia? Please clarify.
Author Response
Thank you so much for your comments
Regarding the broad comments:
It is difficult to summarize the data in tables as the studies are heterogeneous designed. Many previous papers have already presented these informations.
There is no data about comedication with methotrexate
Rituximab is associated to a severe clinical picture as a result of neutropenia and lymphopenia (absolute or functional)
Specific comments
Table 3: do you mean immune mediated neutropenia or auto-immune mediated neutropenia as exclusion criterion? YES
Page 8, lines 232-238: There is a recent study, that shows that common risk factors, such as autoimmune diseases and comedication, may have no impact on neutropenia development for the analgesic dipyrone (doi: 10.1016/j.ejim.2019.07.029 ). It would be interesting to also mention this possibility of drug induced neutropenia. R/ dypirone in auto immune diseases is not associated to an increase risk of neutropenia , thats why we did not mention it
Page 9, lines 269-278: Please give more than one reference for this text passage about pathogenesis. Done
Page 9, line 279: remove “the” before neutropenia Done
Page 9, lines 280-282: the mentioned references do not show the effect of haptens, the mechanism is unknown. Instead, other possible mechanisms could be discussed as reported by Shovman et al. 2015 (Ref. 25). Done
Page 9, lines 282-284: Please give a reference for this information. Done
Page 13, line 455: does antibody cross reactivity of antibiotics need to be considered for every antibiotics induced neutropenia or only for immune mediated neutropenia? Please clarify. Done, we inserted the term immune mediated neutropenia